# From Laboratory towards Industrial Operation: Biomarkers for Acidophilic Metabolic Activity in Bioleaching Systems

**DOI:** 10.3390/genes12040474

**Published:** 2021-03-25

**Authors:** Sabrina Marín, Mayra Cortés, Mauricio Acosta, Karla Delgado, Camila Escuti, Diego Ayma, Cecilia Demergasso

**Affiliations:** 1Centro de Biotecnología, Universidad Católica del Norte, Antofagasta 1240000, Chile; smarin@ucn.cl (S.M.); mayranataliacortesc@gmail.com (M.C.); macostag@ucn.cl (M.A.); kdelgadoteiguel@gmail.com (K.D.); cescuti@ucn.cl (C.E.); 2Departamento de Matemáticas, Facultad de Ciencias, Universidad Católica del Norte, Antofagasta 1240000, Chile; diego.ayma@ce.ucn.cl

**Keywords:** acidophilic bacteria, metabolic monitoring, heap leaching, decision making, relative gene expression, genetic markers, RT-qPCR

## Abstract

In the actual mining scenario, copper bioleaching, mainly raw mined material known as run-of-mine (ROM) copper bioleaching, is the best alternative for the treatment of marginal resources that are not currently considered part of the profitable reserves because of the cost associated with leading technologies in copper extraction. It is foreseen that bioleaching will play a complementary role in either concentration—as it does in Minera Escondida Ltd. (MEL)—or chloride main leaching plants. In that way, it will be possible to maximize mines with installed solvent-extraction and electrowinning capacities that have not been operative since the depletion of their oxide ores. One of the main obstacles for widening bioleaching technology applications is the lack of knowledge about the key events and the attributes of the technology’s critical events at the industrial level and mainly in ROM copper bioleaching industrial operations. It is relevant to assess the bed environment where the bacteria–mineral interaction occurs to learn about the limiting factors determining the leaching rate. Thus, due to inability to accurately determine in-situ key variables, their indirect assessment was evaluated by quantifying microbial metabolic-associated responses. Several candidate marker genes were selected to represent the predominant components of the microbial community inhabiting the industrial heap and the metabolisms involved in microbial responses to changes in the heap environment that affect the process performance. The microbial community’s predominant components were *Acidithiobacillus ferrooxidans*, *At. thiooxidans*, *Leptospirillum ferriphilum*, and *Sulfobacillus* sp. Oxygen reduction, CO_2_ and N_2_ fixation/uptake, iron and sulfur oxidation, and response to osmotic stress were the metabolisms selected regarding research results previously reported in the system. After that, qPCR primers for each candidate gene were designed and validated. The expression profile of the selected genes vs. environmental key variables in pure cultures, column-leaching tests, and the industrial bioleaching heap was defined. We presented the results obtained from the industrial validation of the marker genes selected for assessing CO_2_ and N_2_ availability, osmotic stress response, as well as ferrous iron and sulfur oxidation activity in the bioleaching heap process of MEL. We demonstrated that molecular markers are useful for assessing limiting factors like nutrients and air supply, and the impact of the quality of recycled solutions. We also learned about the attributes of variables like CO_2_, ammonium, and sulfate levels that affect the industrial ROM-scale operation.

## 1. Introduction

The Chilean copper industry will be mostly based on the exploitation of sulfide ore reserves in the coming years [1] because of the exhaustion of oxide resources. Mineral concentration is the primary process currently used for copper extraction from sulfide ores. Then, even when the copper production is expected to increase in Chile, a 50% decrease in the production of electrowon cathodes is estimated. That decrease means a concomitant loss of advantages like a possible reward from the quality of cathodes and additional revenues from the recovery and sale of valuable metals contained in the concentrates. On the other hand, concentrate exports and shipping have potential restrictions and additional costs, and variable charges are also associated with treatment and refining [1]. Nowadays, the development of chemical leaching processes for sulfide ores has gained broad attention in hydrometallurgy due to its faster kinetics and higher recovery when compared to bioleaching [2,3]. Copper leaching in concentrated chloride media under ambient conditions is already used in several Chilean operations to benefit secondary sulfide ore [4]. In contrast, for primary sulfide ores, pilot-scale tests can foresee acceptable copper recovery [3]. However, above all the advantages of the beneficiation processes for sulfide ores mentioned above, the high-energy consumption of crushing and grinding for concentrate production poses an impediment when working with marginal materials. On the other side, one of the main issues in the application of chloride leaching is related to costs and impact on infrastructure due to the necessity of changing metallic equipment in contact with Cl^-^ [4]. In that scenario, copper bioleaching, mainly run-of-mine (ROM) copper bioleaching, is considered the best alternative for the extraction of marginal resources currently not considered part of the profitable reserves because of the cost associated with leading technologies in copper extraction [1]. It is foreseen that bioleaching will play a complementary role in concentration and chloride primary leaching operations, considering that it is not necessary to consider the capital cost for installing the solvent extraction and electrowinning (SX/EW) plants, which are already installed. This was the case at the ROM bioleaching plant at Minera Escondida Ltd. (MEL), where more than 250,000 t of copper as cathodes were produced in 2018, and half of them by bioleaching. Consequently, there is a “clearly crucial need to endow hydrometallurgy with technologies that allow it to increase its efficiency” [1]. The potential of bioleaching to enable recovery of metals from deposits to process available resources that are not amenable to metal recovery by any other means has been assessed, e.g., in Europe as well, aiming to develop environmentally sustainable mining operations [5].

One of the main issues of bioleaching at the mine site is the close control required of acidity, air and nutrients availability, temperature, water quality, and cell activity, among other variables, for better metallurgical performances [6,7,8,9,10,11]. However, it is not feasible to monitor these variables inside ROM industrial permanent heaps, as argued before [12,13]. Furthermore, in terms of mineralogy, the heterogeneous ore bed of sulfide exposure to leachate, solution permeability, and air distribution [14] challenges the control of these variables. ROM permanent heaps have had a remarkable engineering development during the last 20 years. The experience revealed the necessity to solve additional issues, like differential acid feeding, air distribution, and solution recirculation.

The microbial community inhabiting ROM bioleaching heaps at MEL has been monitored based on the relevant taxonomic groups and cell activity determination [15,16]. Following these strategies, we assessed the impact of the microbial community structure on the process’s metallurgical performance [6,15,17]. The microbial diversity was also assessed in other bioleaching operations by using the same strategy [17].

We proposed that a state-of-the-art functional-based strategy to assess habitat-specific functional features should indirectly reveal a heap’s environment conditions, such as bioavailable O_2_, nutrients, and stressful conditions [18,19]. A similar strategy was used to opportunely predict oil contamination in the marine environment [20]. However, up to our knowledge, the analysis of functional genes as biomining community traits was only reported by using cultured microorganisms and in laboratory-controlled conditions [21,22]. Our research work focuses on applying molecular microbial markers for a better control and bioleaching efficiency enhancement.

Based on the genome and metatranscriptomic analyses [9] and the predicted main functions involved in bacterial leaching, we determined the set of genes that better characterizes the microbial response to critical variables in the heap environment. Some of the selected genes were already recognized as the standard for microbial functional ecology [19]. We designed and validated the primers for qPCR, and we performed the experiments at different levels (cultures, column tests, and industrial heap) to validate those genes as biomarkers.

The information obtained from the validated real-time PCR array was incorporated in the development of one of the modules of a decision-support system (DSS) based on decision rules that transform the obtained knowledge into recommendations to the plant operator of the bioleaching process [23,24], assisting with the mine site. This kind of DSS is popularly used, e.g., in evidence-based medicine, to improve healthcare delivery [25]. A clinical decision-support system (CDSS) is a software designed to aid clinical decision-making. In a CDSS, the characteristics of an individual patient are matched to a computerized clinical knowledge base to present the clinician with patient-specific assessments or recommendations. The construction of the DSS for bioleaching processes considers a database for industrial data logging and storage, a knowledge base acquired by suitable statistical and computational tools, and finally, the translation of knowledge into action by applying recommendations that come to terms with operational limitations. The DSS is composed of five reasoning modules: (1) the impact of mineralogy/mineralization in the metallurgical performance, (2) microbial activity and copper recovery, (3) estimated temperature inside the heap, (4) marker genes for critical issues, and (5) inoculation/reinoculation requirements [23].

## 2. Materials and Methods

### 2.1. Strains and Media

*Acidithiobacillus thiooxidans* strain IESL-33 [26] was grown in reactors (3 L) on double-strength (DS) media [27] using sodium tetrathionate (5 mM) as energy source at 30 °C, pH 2.8, with agitation of 200 rpm, and four input CO_2_ concentrations: 50 ppm, 100 ppm, 300 ppm, and 500 ppm (Table 1). The CO_2_ concentration was controlled by bubbling, and input and output gas levels were monitored online using a Micro-Oxymax respirometer (Columbus Instruments, Columbus, Ohio, USA.). Each condition was performed in duplicate.

*Leptospirillum ferriphilum* DSM 14647 type-strain and IESL strain were grown in reactors (1 L) with 600 mL of ABS medium [28], using FeSO_4_ 7H_2_O (50 mM) as an energy source at 37 °C, pH 1.5, with agitation of 200 rpm, and three different concentrations of MgSO_4_ 7H_2_O: 25 mM (control, 2.5 g L^−1^ SO_4_^2−^), 250 mM (25 g L^−1^ SO_4_^2−^), and 500 mM (50 g L^−1^ SO_4_^2^) (Table 1). Each condition was performed in triplicate. The strain DSM 14647 is the type-strain obtained from a culture collection, while IESL is a native strain isolated from the industrial process and maintained in active growth since that. Both *L. ferriphilum* strains have different life histories related to sulfate and probably have different tolerance levels.

*Sulfobacillus* sp. strain CBAR13 (DSM 103670), isolated from MEL industrial bioleaching heap, was grown independently in reactors (1 L) with 600 mL of single-strength [29] and double-strength [27] media, both supplemented with trace elements [28], yeast extract (0.02%), and using FeSO_4_ 7H_2_O (50 mM) and K_2_S_4_O_6_ (2 mM) as an energy source, respectively. Cultures were maintained at 50 °C, pH 1.8, and with agitation of 200 rpm until they reached the late exponential phase. Each condition was performed in triplicate.

Every 3 h during 3 to 4 days, a direct cell count was performed using an improved Neubauer chamber and phase contrast microscope for *L. ferriphilum* and *At. thiooxidans* cultures. Ferrous ion and tetrathionate oxidation activity were monitored by titration with K_2_Cr_2_O_7_ and by potentiometric titration of produced H_2_SO_4,_ respectively [30]. All cultures were harvested in the late exponential growth phase when cells were in abundance and metabolically active. Cells were recovered by vacuum filtration through nitrocellulose membrane (0.2 µm) and were immediately preserved at −80 °C in RNAlater^®^ solution (Invitrogen™ AM7021, Carlsbad, California, USA) until nucleic acid isolation.

We modeled growth (cells·L^−1^) and bacterial activity (Fe^2+^ consumption and H_2_SO_4_ generation) as a function of incubation time and nutrient or stressor concentrations (CO_2_ ppm and SO_4_^2−^, respectively) in order to determine the effect of these variables. Modeling was carried out using a multiple regression model [31] using the Minitab 18 statistical software.

### 2.2. Bioleaching Column Test

Controlled bioleaching experiments run for 105 days at 21–25 °C. Columns (1 m high, 0.6 m diameter) were operated in an open-circuit system (without solution recirculation) and fed with raffinate from the MEL process [32]. Column (col.) tests were performed to assess the effect of air feeding, and NH_4_^+^ and SO_4_^2−^ concentrations in the bioleaching environment (col.1/control, col.2/air-test, col.3/NH_4_^+^-test, and col.4/SO_4_^2−^-test). The columns were inoculated with a microbial consortium from the MEL industrial bioleaching heap. The operation setup was maintained as stable during the test, except for the input-airflow of col.2, which was intentionally interrupted twice by a sustained flow of N_2_ (Table 1), and the NH_4_^+^ and SO_4_^2−^ concentrations of col.3, and col.4, respectively, that were modified twice (Table 1). Samples were collected at different operation times for each microorganism’s quantification and transcriptomic analyses (Table 1). Notwithstanding the relevant metabolic activity previously observed on mineral-associated bacteria [33,34] and its recognized effect on mineral dissolution [35], mineral samples were neither obtained nor analyzed in this work. That decision was taken regarding the research work scope, which was to develop molecular markers for monitoring a mining operation where it is not possible to get representative mineral samples for periodic assessment [12].

Input/output gas (O_2_/CO_2_) concentrations were continuously monitored by a Micro-Oxymax respirometer (Columbus Instruments) in col.2. Ammonia and sulfate levels were daily analyzed by titration after the steam distillation method and gravimetry, in col.3 and col.4, respectively. Cells for relative expression analysis were collected from 40 L of pregnant leaching solution (PLS) by a tangential flow filtration system [9,36] and nitrocellulose membranes of a 0.2 µm pore diameter. Membranes were washed by filtration with 100 mL of TK-444 (0.4 g L^−1^, (NH_4_)2SO_4_; 0.4 g L^−1^, MgSO_4_ × 7H_2_O; 0.4 g L^−1^, K_2_HPO_4_) medium (pH 1.8), 50 mL of sodium citrate (10 mM), and preserved at −80 °C within RNAlater^®^ (Invitrogen™ AM7021) solution until RNA isolation.

### 2.3. Industrial Bioleaching Heap

The bioleaching heap process at MEL considers ROM material that has been characterized as low-grade sulfide ore and heap dimensions of 5000 m long by 2000 m wide [6]. Leaching horizontal sections denominated strips, and five lifts or levels are currently loaded (18 m height each). The strips are irrigated with raffinate from the top of the heap. Raffinate percolates until reaching the heap’s base, enabling dissolved copper from the ore in a PLS. In an attempt to feed the entire ore heap with the O_2_ and CO_2_ necessary for microbial growth and activity, the heap is force-aerated by mechanical blowers located at the base of the heap of each strip. Only some strips of the fourth lift are aerated from their bases as well (Figure 1).

The microbial community at the Escondida sulfide leach process was mainly composed of (i) mesophilic microorganisms, *At. ferridurans* D2, *At. thiooxidans*, and *Ac. acidiphilum*; (ii) thermotolerant microorganisms, *L. ferriphilum* and *At. ferrooxidans* IESL-32; (iii) moderate thermophiles, *At. caldus*, *Firmicutes* representatives from *Sulfobacillus* genus and *Ferroplasma* from the Archaea domain; (iv) extremely thermophilic genus, *Sulfolobus* [6]. Recently, pyrosequencing of 16S rRNA and metagenome sequencing of industrial PLS samples revealed the occurrence of *Thermoplasmatales* archaeon A or E-plasma [37].

Ten PLS samples were collected from five strips (S-132, S-327, S-410, S-413, and S-414) with different aeration conditions and NH_4_^+^ and SO_4_^2−^ levels (Table 2). Briefly, PLS samples (10 L for RNA and 1 L for DNA isolation) from the aerated strip S-410 were obtained biweekly, both at the base of the first lift (drop-D) and the base of the fourth lift (M) of the heap, before (one sample) and after (two samples) an aeration interruption. Aeration suspension was carried out by turning off the aeration blowers (forced aeration) located at the base of the S-410, on the fourth lift (M), while the blowers in the base of the first lift (D) of the strip remained turned on. Besides, PLS samples (10 L for RNA and 1 L for DNA isolation) from the S-132, S-327, S-413 strips (permanently and fully aerated), and from the S-414 strip (permanently and partially aerated, with half of the blowers off) were collected. Operational and physicochemical (pH, Eh, NH_4_^+^, and SO_4_^2−^ concentrations) characterization of samples were performed (Table 2). Cells from the PLS of each strip were harvested, washed, and preserved as described above. Differences in the NH_4_^+^, SO_4_^2−^, and Fe^2+^ concentrations were detected in the strips S-405-D, S-405-B, S-405-M, sampled in 2015, and S-132, S-327, S-413, S-414 sampled in 2017 (Table 2). These samples were used to validate the biomarkers associated with nitrogen metabolism, osmotic stress, and ferrous/sulfur oxidation.

### 2.4. Candidate Genes Selection and Primer Design

To better understand the situation within the heap, it is necessary to know what the microorganisms are doing and how representative they are of the microbial community through gene expression studies and cell quantification. The candidate marker genes were selected to represent the predominant components of the microbial community inhabiting the industrial heap (*At. ferrooxidans*, *At. thiooxidans*, *L. ferriphilum*, and *Sulfobacillus* sp. CBAR13) and the metabolisms involved in microbial responses to changes in the heap environment that affects the process performance. Oxygen reduction, CO_2_ and N_2_ fixation, ammonium transport, iron, and sulfur oxidation, and response to osmotic stress were the metabolisms selected regarding the results of previous research in the system [7,8,10,11].

The metagenomic analysis of a DNA pool generated from industrial samples from the MEL bioleaching heap was previously performed [37]. *At. thiooxidans*, *At. ferrooxidans*, *L. ferriphilum*, and *Sulfobacillus* sp. related sequences were separated through mapping with CLC Genomic Workbench 8.1 (QIAGEN Bioinformatics, Redwood City, CA, USA.) against *Acidithiobacillus*, *Leptospirillum*, and *Sulfobacillus* available genomes. Four representative reconstructed genomes from industrial samples were chosen, respectively, for *At. thiooxidans*, *At. ferrooxidans*, *L. ferriphilum*, and *Sulfobacillus* sp. Reconstruction was performed based on the coverage and the sequencing depth. The sequences of the reconstructed genome were then annotated using the RAST server (https://rast.nmpdr.org/ (accessed on 28 December 2020)) [38].

Primers for RT-qPCR (Appendix A) were designed for protein-coding genes related to CO_2_ fixation, such as *rcbL1*, *rcbL2*, *csoS2*, *csoS3*, *csoS4* from *At. thiooxidans*, and *rcbL1*, *rcbL2* from *At. ferrooxidans* (Table 3). Primers were also designed for protein-coding genes related to synthetic pathways of compatible solutes, nitrogenase activity, ammonium transport and assimilation from *L. ferriphilum*, and protein-coding genes related to the ferrous ion and sulfur oxidation from *Sulfobacillus* sp. CBAR13 (Table 3). Additionally, primers for 16S rRNA, *gyrA*, and *alaS* housekeeping genes (Table 3) were designed for *At. thiooxidans*, *At. ferrooxidans*, *L. ferriphilum*, and *Sulfobacillus* sp. CBAR13 (Appendix A). These housekeeping genes were used as endogenous controls for the relative gene-expression calculation.

DNA sequences of the *At. thiooxidans* selected candidate genes for strains Licanantay (GCA_000709715), ATCC 19377 (GCA_009662475) and the reconstructed Licanantay-like genome from MEL were aligned with ClustalW [39]. Conserved regions were used as the base for the primer design. The same procedure was performed with the candidate genes selected from *At. ferrooxidans*, *L. ferriphilum* strain ML04 (GCA_000299235), SpCl (GCA_001280545), *Sulfobacillus* sp. CBAR13 (LGRO00000000), and the reconstructed genomes from MEL. The promissory candidate primers were evaluated in silico in their specificity with Primer-BLAST (https://www.ncbi.nlm.nih.gov/tools/primer-blast/ (accessed on 28 December 2020)) [40].

All used primers were empirically validated in their specificity, repeatability, and sensibility, as described previously [11]. Pearson’s (R) coefficients of 0.99 and optimal efficiencies (E) (100–110%) [41] were obtained for all selected primers (Appendix A). The annealing temperature for all primers was 60 °C.

### 2.5. RNA Isolation and Transcriptional Analysis

RNA was isolated, and reverse transcription was performed as previously described [9,11].

Transcription dynamics of five *At. thiooxidans* (ATT) Calvin–Benson–Basham (CBB) genes (Table 3) were evaluated in ATT cultures, col.1, col.2 (Table 1), and ten industrial samples (Table 2) by reverse transcription quantitative PCR (RT-qPCR) [6]. Transcription dynamics of two *At. ferrooxidans* (AFE) CBB genes (Table 3) were also evaluated on industrial samples. The AFE–CBB genes were not evaluated in the bioleaching column col.2 because *At. ferrooxidans* did not predominate in that microbial community (results not shown). Since ATT and AFE have the same CO_2_ fixation mechanism (CBB pathway) but different gas requirements, we hypothesize that it is expectable that CBB genes from both species could have a similar response but different expression levels in the industrial environment.

Transcription dynamics of six *L. ferriphilum* (LII) genes associated with the trehalose synthetic pathway for osmotic stress response (Table 3) were evaluated in LII cultures, col.4 (Table 1), and industrial strips S-132, S-327, S-413, and S-414 (Table 2). Three genes associated with nitrogen metabolism (Table 3) were evaluated in col.3 and industrial strips S-132, S-327, S-413, S-414, and S-405 (Table 2). Each qPCR run was performed using technical triplicates.

Transcription dynamics of four genes associated with ferrous oxidation and two genes associated with sulfur oxidation (Table 3) were evaluated in *Sulfobacillus* sp. CBAR13 (STO) pure cultures (Table 1) and industrial strips S-132, S-327, S-413, and S-414 (Table 2).

The fold-change gene expression (ratio) was calculated by the Pfaffl model using REST software V2.0.13 [42]. The Pfaffl model has been successfully used in previous gene expression analyses for acidophilic microorganisms [6,9,11]. One sample of each sample-set was selected and used as a control sample. Relative gene expression was calculated as the transcription level in target samples relative to the control sample. All relative gene expression analyses were normalized against endogenous controls previously validated by the NormFinder algorithm [43]. Relative expression levels were expressed as Log_2_-ratio. Significant differences in relative expression were adjusted at −1 < Log_2_ ratio < 1.

## 3. Results

### 3.1. Cell Harvesting, DNA, and RNA Extraction

RNA quantity obtained from culture samples ranged between 20–200 ng/µL while the quality for 260/280 and 260/230 ratios ranged between 1.5–2.1 and 0.9–2.0, respectively, as expected for this kind of sample. The analysis performed on the RNA obtained from bioleaching columns and industrial-strip samples (Appendix A) revealed that, in general, RNA presented good and acceptable quantity and quality ranges for transcriptomic RT-qPCR analyses.

### 3.2. Biomarkers for CO_2_ Availability in Acidithiobacillus spp.

#### 3.2.1. Effects of CO_2_ Availability on Growth and Activity of *At. thiooxidans* Strain IESL-33

Different CO_2_ concentrations in all four reactors were maintained as stable through time in the experiments (Figure 2). The growth rates of *At. thiooxidans* IESL-33 as assessed by cell counts was slower at lower CO_2_ concentrations. Significant differences (*p* < 0.05) (Appendix A) were observed between reactors operated at 50 and 500 ppm of CO_2_ (Figure 2A). No significant differences were observed in growth between reactors operated at 50 and 100 ppm of CO_2_ or reactors operated at 300 and 500 ppm of CO_2_. The microbial activity expressed as acid generation (H_2_SO_4_)—the product of bacterial sulfur oxidation activity—showed a lower rate of production in reactors with lower CO_2_ availability (Figure 2B).

#### 3.2.2. Identification of Transcriptomic Markers Associated with CO_2_ Availability

Culture batch reactors of *At. thiooxidans* (ATT) IESL-33 at 50 ppm, 100 ppm, and 500 ppm of CO_2_ showed an over-expression of ATT-*rbcL*-*1*, ATT-*csoS2*, ATT-*csoS3*, and ATT-*csoS4B* genes, and an under-expression of ATT-*rbcL*-*2* gene relative to the control condition (300 ppm CO_2_) (Figure 3A). The highest over-expression of these genes occurred in the culture growing with 50 ppm of CO_2_ (Figure 3A), suggesting an apparent transcriptomic response to low CO_2_ availability. The ATT-*csoS3* gene associated with carbonic anhydrase [44] was significantly over-expressed only at relatively low CO_2_ levels (Figure 3A), as was also observed in the column test (Figure 3B).

In the column test (col.2), non-significant differences were observed in the expression level of the ATT-*rbcL*-*1* gene among samples with relatively low CO_2_ levels (0–500 ppm CO_2_-out). Meanwhile, this gene’s under-expression was evident in samples with relatively high CO_2_ levels (1500–2000 ppm CO_2_-out) (Figure 3B). Besides this, the over-expression of the ATT-*rbcL*-*2* gene was evidenced at day 64 with 2000 ppm CO_2_-out.

The transcription dynamics of the ATT-*csoS3* gene were similar to the ATT-*rbcL*-*1* gene, except after the second decrease of CO_2_ on day 76 (200 ppm CO_2_-out) when ATT-*csoS3* was over-expressed (Figure 3B).

#### 3.2.3. Industrial Validation of Transcriptomic Markers Associated with CO_2_ Fixation

Transcriptional analyses were performed using industrial strips sampled for aeration (Table 2) because the cDNA of strips S-410-D1, S-410-M2, S-413-D did not show PCR amplification for any of the candidate marker genes. This result was possibly due to the absence of the target microorganisms or the absence of the expression of those genes. Besides this, some genes like ATT-*csoS2*, ATT-*csoS3*, and *rbcL2* from *At. ferrooxidans* (AFE) did not show amplification in any examined industrial samples (Figure 4). The absence of significant differences (Log_2_ ratio < 1) in the relative expression level of the ATT-*rbcL*-*1* gene among S-410-M0, S-410-D4, and S-410-M4 (Figure 4A) suggested that similar levels of CO_2_ were available in these samples. Only the PLS solution from S-410-D4, obtained from the drop four weeks after aeration interruption, showed a slight gene over-expression, which allows us to infer a relatively low CO_2_ availability for *At. thiooxidans*. A different response was observed in *At. ferrooxidans* AFE-*rbcL*-*1* gene, which was slightly over-expressed in S-410-D2 (obtained from the drop two weeks after aeration interruption), and no expression differences were observed in the rest of the samples (Figure 4A). The ATT-*rbcL*-*2* gene was over-expressed in samples S-410-D2 and S-410-M4 (Figure 4A) and partially aerated strip S-414-D (Figure 4B).

### 3.3. Biomarkers for N_2_ Availability in L. ferriphilum

#### 3.3.1. Identification of Transcriptomic Markers Associated with N_2_ Availability

Transcriptional analyses were performed with PLS samples from the column test (col.3), where NH_4_^+^ concentration was modified. The relative gene expression of *nifH*, *amt*, *glnA*, and *glnB* genes from *L. ferriphilum* were determined using two validated reference genes (16S rRNA and *alaS*).

Lower expressions of *nifH* (involved in dinitrogen fixation, Table 3), *glnA*, and *glnB* (involved in the assimilation of ammonium, Table 3) were detected in most of the samples when the NH_4_^+^ concentration was increased in the bioleaching column col.3 (Figure 5). The transcriptional pattern observed for *glnA* and *glnB* genes was different in the first sample (operation day 76) after the increment of NH_4_^+^ (day 62). On the contrary, the *amt* gene (involved in the transport of ammonium, Table 3) was only under-expressed when a high level of NH_4_^+^ was present in col.3 (Figure 5). All genes were over-expressed on day 104, 13 days after the second interruption of the NH_4_^+^ supply in the column.

#### 3.3.2. Industrial Validation of Transcriptomic Markers Associated with N_2_ Metabolism

Transcriptional analyses for industrial validation of biomarkers associated with nitrogen fixation were performed using sampled strips (Table 2). Some genes like *glnA* did not show amplification in any of the industrial samples. The relative gene expression was analyzed using two validated reference genes (16S rRNA and *alaS*).

An increase in the expression of *nifH* was observed when the NH_4_^+^ level decreased to 3 mg L^−1^ in the industrial leaching solutions analyzed (Figure 6). The pattern observed in the *amt2* and *glnA* gene’s transcription does not relate to the registered concentration of NH_4_^+^. It is crucial to consider that the level of NH_4_^+^ reached in the columns was higher than the high level reported from the industrial solutions.

### 3.4. Biomarkers for Osmotic Stress in L. ferriphilum

#### 3.4.1. Effect of SO_4_^2−^ Concentration on Growth and Activity of *L. ferriphilum* Strains DSMZ 14647 and IESL-25

The *L. ferriphilum* strain DSM 14647 growth rate was significantly lower (*p* < 0.0001) at 25 and 50 g L^−1^ SO_4_^2−^ than at 2.5 g L^−1^ (Figure 7A) (Appendix A). However, there was not enough data to confirm the significance of the decrease in the oxidation activity measured by the Fe^2+^ consumption at higher SO_4_^2−^ concentrations (Figure 7B) (Appendix A).

The native *L. ferriphilum* strain IESL-25 growth rate was significantly lower (*p* < 0.0001) at 25 and 50 g L^−1^ SO_4_^2−^ than at 2.5 g L^−1^ SO_4_^2−^ (Figure 8A). The negative effect of high SO_4_^2−^ levels was evidenced by the significant decrease (*p* < 0.05) in the rate of Fe^2+^ consumption in cultures growing with 50 g L^−1^ SO_4_^2−^ (Figure 8B) (Appendix A).

#### 3.4.2. Identification of Transcriptomic Markers Associated with Osmotic Stress

Transcriptional analyses were performed with pure culture samples (three conditions on triplicate) of *L. ferriphilum* DSMZ 14647 growing under different SO_4_^2−^ concentration and PLS samples from the column test (col.4) where SO_4_^2−^ concentration was modified. The relative gene expression of *treZ*, *treY*, *lamb*, *treX*, *galU*, and *gadB* genes was analyzed using two validated reference genes (16S rRNA and *alaS*).

The transcriptional dynamic of the selected genes involved in the trehalose synthesis pathway in the reactor culture of *L. ferriphilum* type strain DSM 14647 showed an increase at high SO_4_^2−^ levels (Figure 9A) relative to the control sample (2 g L^−1^ SO_4_^2−^). The transcription of the *gadB* gene only increased when culturing with 50 g L^−1^ SO_4_ (Figure 9A). Non-significant gene transcription differences were observed between the triplicates of *L. ferriphilum* DSMZ 14647 control cultures growing on 2.5 g L^−1^ SO_4_ or between the different analyzed genes in the pure culture tests (Figure 9A).

Four of the six genes associated with compatible solutes in *L. ferriphilum* were quantified in the bioleaching column test (col.4) after the first and the second increment of sulfate concentration in the feeding solution (Figure 9B) and relative to the non-intervened sample used as control (day 67). The over-expression of *treYZ* genes remained stable during both sulfate interventions performed on operation days 68 and 103, respectively (Figure 9B). The *lamB* and *galU* genes were only over-expressed on day 91, after 23 days since the first sulfate increment from 50 to 80 g L^−1^. The gene of the trehalose biosynthetic pathway OtsAB (I) represented by the gene *galU* maintains its expression level despite sulfate increase. A similar result was observed in the expression level of the *lamB* gene related to trehalose transport (Figure 9B).

#### 3.4.3. Industrial Validation of Transcriptomic Markers Associated with Osmotic Stress

Transcriptional analyses for industrial validation of biomarkers associated with osmotic stress were performed using the sampled strips (Table 2). Some genes like *treX*, *gadB*, and *galU* did not show amplification in the industrial samples. The relative gene expression was analyzed using the two validated reference genes (16S rRNA and *alaS*).

The *treZ*, *treY*, and *lamB* genes from *L. ferriphilum* were over-expressed in industrial PLS samples related to high SO_4_^2−^ concentrations (Figure 10A,B). The couple *treYZ* was over-expressed either in industrial strips obtained during the same sampling activity (Figure 10A) as in strips obtained during independent samplings (Figure 10B) with concentrations over 90 g L^−1^ SO_4_^2^.

### 3.5. Biomarkers for Iron and Sulfur Oxidation in Sulfobacillus sp. CBAR13

#### 3.5.1. Effect of the Energy Source on Growth of *Sulfobacillus* sp.

The *Sulfobacillus* sp. strain CBAR13 growth rate was significantly higher on SS medium with iron as the energy source than under DS medium with tetrathionate (Figure 11).

#### 3.5.2. Identification of Transcriptomic Markers Associated with Ferrous and Sulfur Oxidation

Transcriptional analyses were performed with pure culture samples (two conditions on triplicate) of *Sulfobacillus* sp. CBAR13 growing with different electron donors (Fe^2+^ and sulfur). The relative gene expression of *cytbdl*, *sulf*, *chypII*, *carb*, *tehy*, and *fadp* genes (Table 2) were analyzed using the validated reference gene *gyrA*.

Transcriptional dynamics of the *cytbdl*, *sulf*, *chypII*, and *carb* genes from culture batch reactor samples of *Sulfobacillus* sp. CBAR13 growing on SS medium (50 mM FeSO_4_ 7H_2_O) showed a significant over-expression from one to six-folds relative to one of the replicates growing on DS medium (2 mM K_2_S_4_O_6_) (Figure 12A). Besides which, a significant under-expression from five to nine-folds for *tehy* and *fadp* genes associated with tetrathionate oxidation (Table 3) was observed in cultures growing on SS medium (Figure 12A). Nonsignificant transcriptional differences were observed between the triplicates of *Sulfobacillus* sp. CBAR13 growing on DS cultures (Figure 12A).

#### 3.5.3. Industrial Validation of Transcriptomic Markers Associated with Ferrous and Sulfur Oxidation

Transcriptional analyses for industrial validation of biomarkers associated with ferrous/sulfur oxidation (Table 2) showed a significant increment near six-folds on the gene *tehy* transcription in strip S-327 with 0.04 g L^−1^ Fe^2+^ relative to strip S-132 with 0.24 g L^−1^ Fe^2+^ (Figure 12B and Table 2). Inversely, an under-expression was observed for *sulf* and *chypll* genes in the industrial sample with less Fe^2+^ availability (S-327) (Figure 12B). The relative gene expression was analyzed using the validated reference *gyrA* gene.

The analyses were performed using the sampled strips (Table 2). PCR amplification was not obtained from cDNA of strips S-413 and S-414 for any candidate marker genes. Genes *cytbdl*, *carb*, and *fadp* did not show amplification in the industrial samples (Figure 12B).

## 4. Discussion

The results obtained for validating genetic markers associated with O_2_ availability within the heap were previously reported [11].

### 4.1. Carbon Fixation

It is known that bioleaching processes rely on O_2_ to promote the oxidation processes for sulfide mineral dissolution and on CO_2_ as the carbon source for the growth of microorganisms, which catalyze the oxidation. An important issue in heap bioleaching technology is to install homogeneous aeration for providing enough O_2_ and CO_2_ in the environment to improve operational performance. The pH, temperature, ion concentration, carbonate content in the mineral, altitude, and activity of heterotrophic microorganisms also influence the CO_2_ availability [28]. A significant decline of microbial oxidation rates was evidenced in CO_2_ depleted zones in a controlled heap-like environment, sealed large columns operated isothermally at 40 °C and supplemented with additional CO_2_ [45]. Besides, the effect of CO_2_ was also determined in agitated reactors, and increased copper extraction was observed when CO_2_ was supplied at 4% (4000 ppm) [46].

The results obtained from reactor experiments of *At. thiooxidans* growing at different CO_2_ concentrations (Figure 2A,B) confirm previous studies regarding the necessity of elevated CO_2_ concentrations for optimum growth in *At. thiooxidans* [47]. On the contrary, it has been reported that the closely related species *At. ferrooxidans* that use the same CBB cycle reach an optimum specific growth rate (*µ*) at 100 ppm of CO_2_ [28]. However, *At. thiooxidans* maintains an increasing *µ* tendency until 3000 ppm of CO_2_ [47] while a decline of *µ* was registered in *At. ferrooxidans* growing at CO_2_ concentration above 100 ppm [28].

The characterization of the microbial community composition after air feeding interruption in col.2 evidenced a decrease of acidithiobacilli and *L. ferriphilum* and a similar abundance, or even an increase, of heterotrophic microorganisms like *Thermoplasma* sp. and *Ferroplasma acidiphillum* (Appendix A). Then, the increased concentration of CO_2_ on the output of col.2 (Figure 3B) was probably due to heterotrophic microbial metabolism. In this experiment, the air was displaced with an equal flux of N_2_, so the gas-in mass input changes were discarded.

Previous works have shown that the growth of *At. thiooxidans* is favored by an elevated concentration of CO_2_ (2500 ppm) [47]. Our results show that ATT *rbcL*-*1* transcription was associated with relatively low CO_2_ availability (Figure 3A,B). Similar results in the transcription of *rbcl-1* have been reported for *At. ferrooxidans* [9]. The over-expression of ATT *rcbL-2* observed at the first airflow interruption in col.2 (Figure 3B) suggests that the transcription of this gene could be associated with relatively high CO_2_ levels, as it has been previously reported for other species [48]. It is generally concluded that RubisCo type II enzymes are adapted to function at low-O_2_ and medium to high-CO_2_ environmental levels [49], a feature supported by these results.

Apparently, relatively low CO_2_ levels induced an increased transcription of ATT *rbcL*-*1* and ATT *csoS3* genes, as we have previously reported for acidithiobacilli [9]. The structural carboxysome *csoS2* and *csoS4B* genes were over-expressed at both high and low CO_2_ levels. A plausible explanation could be the differences in the turnover between the structural (e.g., CsoS2 and CsoS4B) and functional proteins (e.g., CsoS3), or it indicates a CO_2_-storage function in carboxysome. More evidence is required to confirm these hypotheses.

The transcriptional dynamics observed in the industrial samples (Figure 4A,B) support previous findings that *At. thiooxidans* and *At. ferrooxidans* have different strategies to respond to changes in CO_2_ levels, although both species share the same CO_2_ uptake/fixation mechanism [50].

The over-expression of the ATT *rbcL*-*2* gene (Figure 4A,B) is probably suggesting a response of *At. thiooxidans* to high CO_2_ and low O_2_ levels as has been recognized for this gene [49] and observed in pure cultures and bioleaching column tests as well (Figure 3A,B).

The consistent results of transcriptional dynamics obtained from the pure culture at the laboratory level, controlled column tests, and industrial samples allowed us to obtain real indicators for CO_2_ availability in bioleaching systems. The industrial application of those indicators in monitoring forced aeration in a heap could influence the control and design of present and future bioleaching operations, respectively.

Finally, we show that the growth and the oxidation activity of *At. thiooxidans* IESL-33 was slower when the cells were exposed to a low concentration of CO_2_, also accompanied by a lower metabolic activity (Figure 2 and Appendix A), confirming the results of previous works [28]. Furthermore, we found that changes in the airflow input in bioleaching column tests and the MEL heap’s industrial operation influence the microbial population (Appendix A). These changes can be related to metabolic shifts that microorganisms must overcome during periods of low oxygen concentration. Finally, the transcriptional analysis of the biomarkers for CO_2_ metabolism of *At. thiooxidans*, *rbcL*-*1* and *rbcL*-*2* showed a similar response to that observed in *At. ferrooxidans* but at different CO_2_ or O_2_ concentrations.

### 4.2. Nitrogen Metabolism

The ore in a heap provides the micronutrients required for the microbial community’s activity responsible for sulfide minerals dissolution. However, a supply of macronutrients, such as ammonium, potassium, and phosphate could be needed [51]. There is, as well, an input of ammonium in the system, caused by the ability of some microorganisms like *At. ferrooxidans* and some members of *Leptospirillum* genus to fix nitrogen from the atmosphere [8,51,52].

The ammonium concentration measured at the heap is on average 5.5 mg L^−1^, and higher levels (up to 10 mg L^−1^) were observed after the start of irrigation of some new strips (Appendix A). Similar levels were reported in leaching solutions on a heap bioleaching process from Talvivaara Mining Company [51]. Over-expression of genes involved in the nitrogen fixation (*nifH*) was observed in the column test when NH_4_^+^ concentration decreased up to 3 mg L^−1^, and a similar behavior was observed in the industrial leaching solutions analyzed (Figure 6).

Previous results have shown a significant increase of the *nifH* expression level after 110 days of heap operation and an opposite profile for the *amt* gene of *L. ferriphilum*. Even though we have not measured the ammonium concentration in that opportunity, the results insinuated a higher availability of ammonium at the initial stages of the process and a metabolic shift to N_2_ fixation when the operation time progressed. The results from ammonium analysis of randomly selected strip PLSs confirmed the occurrence of ammonium levels above the average during the initial irrigation of some strips from the first lift (Appendix A).

A high diversity of *L. ferriphillum* was observed in the industrial heap with a predominance of sequences from Group IIc [6,37]. Group IIc is composed of strains with and without the capacity for N_2_ fixation [8,52,53,54]. The N_2_ fixation capacity was confirmed in the IESL-25 strain [54] isolated from the industrial heap [55]. Different gene localization was observed in the nitrogen fixation operon of the pan-genome of *Leptospirillum* spp. retrieved from the heap [37].

### 4.3. Osmotic Stress

The primary response of bacteria to exposure to a high osmotic environment is the accumulation of specific solutes like K^+^, glutamate, γ-aminobutyrate, trehalose, proline, and glycine betaine at concentrations that are proportional to the osmolarity of the medium [56].

Trehalose is a sugar that has been described as the compatible solute mainly used by *L. ferriphilum* in response to osmotic stress. Increments of trehalose synthesis until 59 folds after exposition to high concentrations of MgSO_4_ have been previously reported [57]. Genes involved in the three trehalose biosynthetic pathways: GalU-OtsAOtsB (I), TreY-TreZ (V), and TreS (IV) [58,59], were identified in the genome of *L. ferriphilum* Sp-Cl [54]. The transcription dynamics of genes involved in the TreYZ pathway observed in this study from pure cultures, controlled bioleaching columns, and industrial bioleaching heap (Figure 9 and Figure 10) confirms that *L. ferriphilum* prefers the TreYZ pathway under osmotic stress triggered by high SO_4_^2−^ concentrations.

The over-expression of *treYZ* genes observed in all the mentioned scenarios (Figure 9 and Figure 10) confirms the participation of the TreYZ pathway in the response to osmotic stress in all the analyzed sulfate concentrations in *L. ferriphilum* and the *treYZ* genes as industrially validated transcriptomic markers for the monitoring of the osmotic stress in bioleaching systems.

The UDP-glucose pyrophosphorylase enzyme coded by the *galU1* gene [60] catalyzes a reaction between the glucose-1-phosphate and the UTP to form UDP-glucose. UDP-glucose participates in glycogen synthesis, which, in turn, can be used as a precursor for the trehalose synthesis by the TreYZ pathway [61,62]. Previous studies report that the simultaneous over-expression of *galU*, *otsAB*, and *treYZ* genes triggers a six-fold increment in the trehalose production in *Corynebacterium glutamicum* [60,61]. The results of the present study allow us to infer that *L. ferriphilum* type-strain (DSMZ 14647) and *L. ferriphilum* “wild” strains from the industrial bioleaching heap use the TreYZ pathway to increase the trehalose synthesis under prolonged conditions of osmotic stress (Figure 9A,B).

The *lamB* gene expression is associated with the trehalose transport from the cytoplasm to the cell periplasm. It can then be degraded to two glucose molecules by a trehalase (TreA) [63]. The over-expression of the *lamB* gene observed at all the analyzed sulfate concentrations (Figure 9A,B and Figure 10A) confirms that this gene is also transcribed by *L. ferriphilum* in response to osmotic stress (Figure 9 and Figure 10). It is possible to infer that *L. ferriphilum* can regulate the intracellular concentration of solutes by *lamB* gene transcription when trehalose synthesis has been increased by the co-transcription of *treYZ*/*galU* genes.

The transcription of the *gadAB* genes is associated with the synthesis of the glutamate decarboxylase, a pyridoxal 5′-phosphate-dependent enzyme that catalyzes the glutamate conversion producing λ-aminobutyric acid (GABA) by an irreversible decarboxylation reaction [64]. From the *gadB* transcription-dynamic results observed only in the pure culture tests (Figure 9A), it is reasonable to assume that glutamate is probably not the compatible solute preferred by *L. ferriphilum* under osmotic stress conditions. This issue agrees with previously reported results confirming trehalose as the preferred compatible solute in *L. ferriphilum* [57]. The *gad* gene does not participate in trehalose synthesis, but it has been reported that the enzyme is required for trehalose protection against acid stress. The over-expression of both genes simultaneously could indicate the occurrence of acid stress [65].

The congruent transcription dynamic observed from *treYZ* and *lamB* genes on the pure cultures, bioleaching-column tests, and industrial bioleaching heap certainly allow us to validate these three genes as transcriptional markers associated with osmotic stress response in *L. ferriphilum*. The *treYZ* and *lamB* genes can be useful for eventual metabolic monitoring of industrial copper-bioleaching processes.

### 4.4. Ferrous and Sulfur Oxidation

The candidate genes associated with ferrous (*cytbdI*, *sulf*, *chypII*, *carb*) and sulfur (*tehy*, *fadp*) oxidation pathways (Table 3) were over-expressed and under-expressed, respectively, in pure cultures of *Sulfobacillus* sp. CBAR13 growing in the presence of iron as the only energy source (SS) relative to cultures growing with tetrathionate as electrons donor (DS) (Figure 12A). The same transcriptional dynamics were observed for *sulf*, *chypII*, and *tehy*-genes analyzed on the industrial samples (Figure 12B), validating those genes as molecular markers for monitoring iron and sulfur oxidation in industrial heap-bioleaching systems. The *cytbdI*, *carb*, and *fadp* genes did not show amplification in the analyzed industrial samples, so they were proposed as putative molecular markers for the energy generation metabolism in *Sulfobacillus* until the industrial validation be reached. The absence of amplification could be due to the low sensibility of the primers. New primers will be designed and assayed for these three genes in future experiments.

The results observed agreement with the previously reported substrate behavior in tetrathionate-adapted and iron (II)-adapted cells of three species of *Sulfobacillus* genus. In that research, the authors concluded that a ‘preference’ for ferrous oxidation appears to be expected in this genus. The requirement of ferric ions for growth on tetrathionate would explain that feature [66].

While a ‘preference’ for ferrous oxidation appears to be shared in this genus, understanding how the cells convert the chemical energy to biomass will require significant further study.

*Sulfobacillus* genomes were predicted to possess tetrathionate hydrolase (*tehy*), which directs the disproportion of tetrathionate to generate sulfate, thiosulfate, and sulfur [67,68]. In this work, the *tehy* gene was under-expressed near ten-folds in pure cultures growing with ferrous ion as the electron donor (SS) (Figure 12A) and over-expressed near six-folds in the industrial strip with a lower Fe^2+^ availability (S-327) (Figure 12B, Table 2). Similar transcriptional dynamics were observed in *S. acidophilus*, where the *tehy* gene was over-transcribed 1.94-folds and under-transcribed 0.47-folds in cultures grown in the presence of sulfur and FeSO_4_, respectively [69].

Blue-copper proteins are members of the cupredoxin superfamily and have been found related to aerobic ferrous iron oxidation in bioleaching systems (e.g., [70]). The *sulf* gene related to sulfocyanin blue-copper protein from *Sulfobacillus* was over-expressed in the CBAR13 strain growing on SS medium with Fe^2+^ and under-expressed in the industrial strip with lower Fe^2+^ availability (Figure 12A,B). The transcription dynamics observed in the present study support the RNAseq [71], and the semi-quantitative RT-PCR [67] results showed previously. Both studies revealed an over-expression of putative genes associated with blue-copper-like proteins (sulfocyanin/rusticyanin) from *S. thermosulfidooxidans* on a chalcopyrite bioleaching column test when *Sulfobacillus* proportion in the column was >80% and the Fe^2+^ oxidation rate measured at 45 °C reached 10 mg L^−1^h^−1^, and during growth on ferrous sulfate, respectively.

The *sulf* and *chypII* genes associated with the sulfocyanin protein and the *cbb3*-type cytochrome-c-oxidase-subunit-I, respectively, and both putatively related to iron oxidation in *Sulfobacillus* sp. CBAR13 were under-expressed in industrial strip S-327 (0.04 g L^−1^ Fe^2+^) relative to S-132 (0.24 g L^−1^ Fe^2+^) (Figure 12B). However, the level of under-expression of those genes observed in the industrial strip with lower Fe^2+^ availability was less evident than the over-expression of the ferrous ion oxidation genes evidenced in the *Sulfobacillus* sp. CBAR13 pure cultures growing with Fe^2+^ as the energy source (Figure 12). The stronger iron-oxidizer *L. ferriphilum* present could explain this result in the studied industrial MEL strips (Appendix A). It has been reported that some electron chain components, linked to iron oxidation in *Sulfobacillus* sp. CBAR13, are down-regulated in the presence of a second iron oxidizer as *L. ferriphilum* [72]. The above is probably because the concentration of available iron is too low for utilization by *Sulfobacillus* sp. CBAR13. Meanwhile, *L. ferriphilum* can scavenge Fe^2+^ at concentrations far below *Sulfobacillus* sp. CBAR13 capabilities but that *L. ferriphilum* is still able to use [72].

A set of 16 marker genes industrially validated throughout our work for O_2_, CO_2_, and N_2_ availability, osmotic stress, and sulfur/ferrous oxidation study in *Acidithiobacillus* sp, *Leptospirillum* sp., and *Sulfobacillus* sp., conform the first qPCR-array for monitoring critical metabolic status in copper industrial bioleaching systems (Figure 13). Furthermore, the rules constructed based on these results and already incorporated in the DSS, opportunely advise the operator about deviating parameters, that cannot be directly determined in the heap, from the optimal range for the microbial process.

## 5. Conclusions

The array of transcriptomic markers developed and validated in this study strongly represents the metabolic scenario involved in microbial responses to parameters that influence the industrial bioleaching heap process’s productivity and efficiency. The transcriptomic dynamics resulting from the industrial strips analyses evidence of an active response to CO_2_ and N_2_ limitation, osmotic stress, and active iron and sulfur microbial oxidation.

Molecular markers were demonstrated to be useful for indirectly assessing critical environmental variables (events), which are known as limiting factors for the rate of copper leaching in bacterial-assisted plants. The results have allowed us to provide new data on parameters that should be improved to optimize bioleaching, such as CO_2_ and NH_4_^+^ concentrations. Moreover, this array, applied for periodically assessing the industrial bioleaching operations, will provide the amount of data needed to get the industry knowledge required to encourage innovation. The incorporation of those results in the DSS, by means of rules, transfers the knowledge acquired to the operator.

The possibility of transferring this approach to other bioleaching plants is high because the microbial community structure inhabiting the bioleaching processes is mostly a traceable one, where *Acidithiobacillus*, *Leptospirillum*, and *Sulfobacillus* are the predominant members.

## Figures and Tables

**Figure 1 genes-12-00474-f001:**
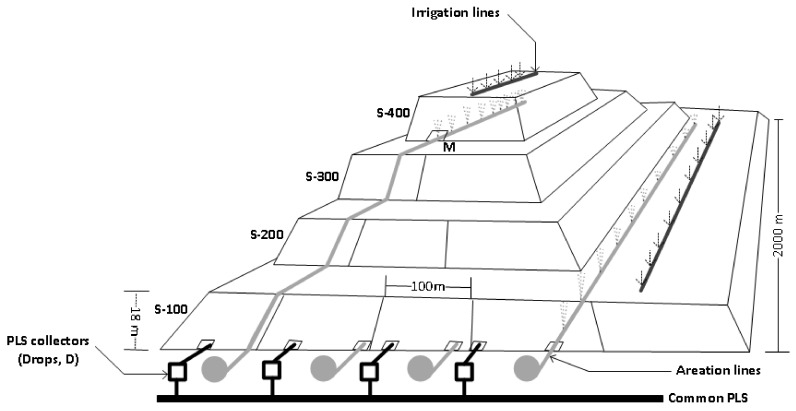
Schematic drawing of the bioleaching heap at Escondida mine with four lifts. D corresponds to drops. M is the base of the fourth lift (adapted from [11]).

**Figure 2 genes-12-00474-f002:**
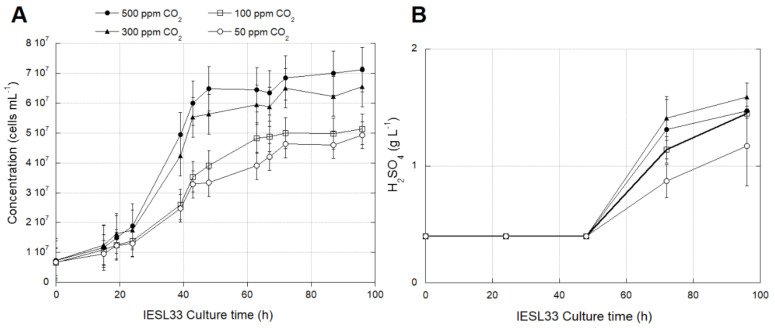
Cell growth (**A**) and acid production (**B**) of *At. thiooxidans* IESL-33 cultured under different CO_2_ concentrations. Vertical bars represent the standard deviation (SD) calculated with biological replicates.

**Figure 3 genes-12-00474-f003:**
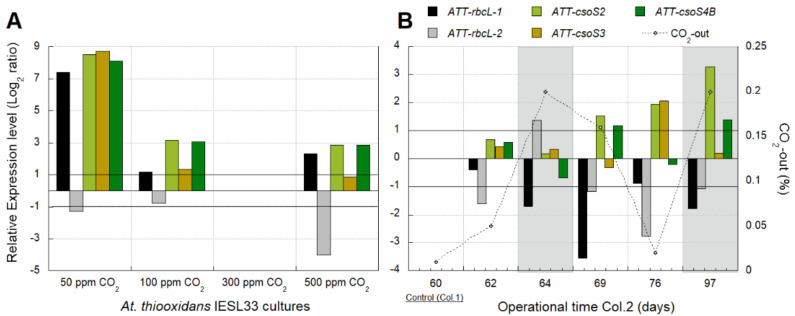
Transcriptional dynamics of five key Calvin–Benson–Basham (CBB) genes of *At. thiooxidans* (ATT) in response to (**A**) four different CO_2_ supply conditions in pure cultures of the strain IESL-33 and to (**B**) variations in the CO_2_ availability produced by three different interventions of input airflow in a controlled bioleaching column. Gray areas in the second graph show the two periods without input airflow in col.2 (days 63 to 68 and days 91 to 104). Significant differences in relative expression were adjusted at −1 < Log_2_ ratio < 1 (horizontal black lines). Gene expression was calculated relative to the control samples (log 0 values) that correspond in (**A**,**B**), respectively, to the culture with 300 ppm and the sample from operational day 60. In (**B**), the X-axis corresponds to operation days at the sampling time of col.2.

**Figure 4 genes-12-00474-f004:**
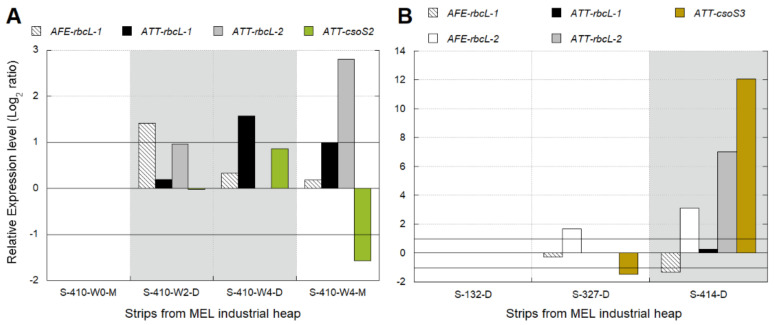
Transcriptional dynamics of key CBB-genes of *At. thiooxidans* (ATT) and *At. ferrooxidans* (AFE) in response to (**A**) industrial aeration interventions performed in strip S-410 and (**B**) in strips S-132, S-327, and S-414. In (**A**) 0, 2, and 4 are the three samplings performed biweekly before (0) and after (2, 4) aeration intervention in strip S-410 (Table 2). The samples D were collected from the drop located at the base of the heap. The samples M were collected at the base of the fourth lift of the heap. Gray areas show the strips without aeration (blowers turned off) in the base of the fourth lift. Significant differences in relative expression were adjusted at −1 < Log_2_ ratio < 1 (horizontal black lines). Gene expression was calculated relative to the control samples (log 0 values) that correspond in (**A**,**B**), respectively, to S-410-W0-M and S-414-D.

**Figure 5 genes-12-00474-f005:**
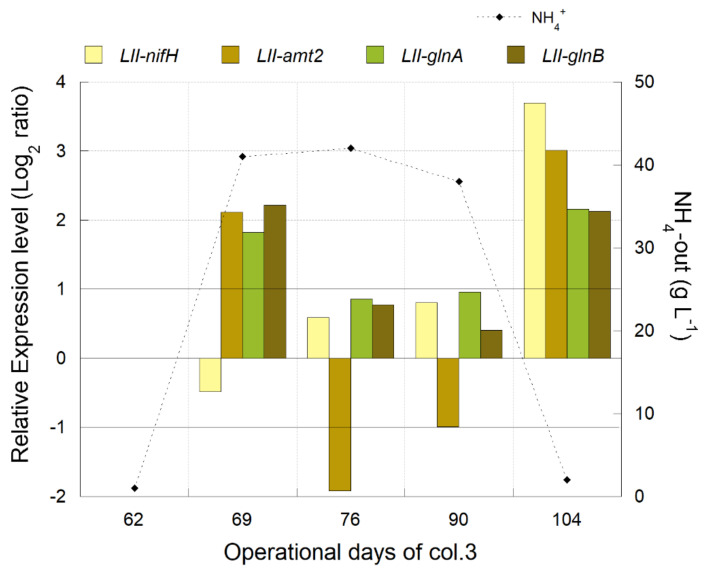
Transcriptional dynamics of *nifH*, *amt2*, *glnA*, and *glnB* genes of *L. ferriphilum* (LII) during NH_4_^+^ interventions (NH_4_^+^ supplemented on and off at operation days 62 and 90, respectively) in the bioleaching column, col.3. Significant differences in relative expression were adjusted at −1 < Log_2_ ratio < 1 (horizontal black lines). Gene expression was calculated relative to the control sample (log 0 values) that corresponds to operational day 61. The *X*-axis corresponds to operation days at the sampling time of col.3.

**Figure 6 genes-12-00474-f006:**
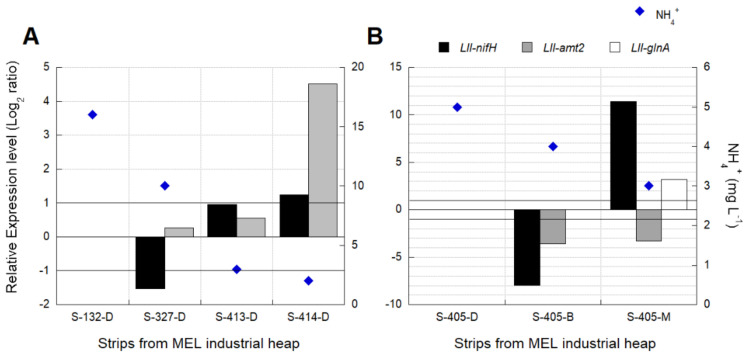
Transcriptional dynamics of *nifH*, *amt2*, and *glnA* genes in *L. ferriphilum* (LII) from (**A**) strips S-132, S-327, S-413, S-414, and (**B**) S-405 of the MEL industrial copper bioleaching heap with different NH_4_^+^ availabilities. D = Drop or base of the heap, B = the base of the second lift, M = the base of the fourth lift. Significant differences in relative expression were adjusted at −1 < Log_2_ ratio < 1 (horizontal black lines). Gene expression was calculated relative to the control samples (log 0 values) that correspond in (**A**,**B**), respectively, to S-132 and S.405-D.

**Figure 7 genes-12-00474-f007:**
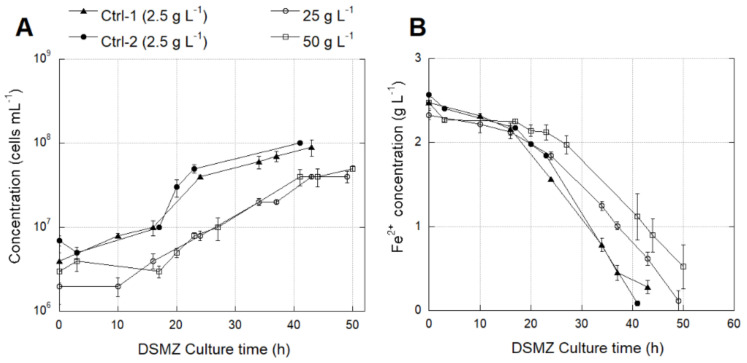
Cell growth (**A**) and Fe^2+^ consumption (**B**) of *L. ferriphilum* type strain DSM 14647 cultured under different SO_4_^2−^ concentrations (g L^−1^). Vertical bars represent the standard deviation (SD) calculated with biological replicates.

**Figure 8 genes-12-00474-f008:**
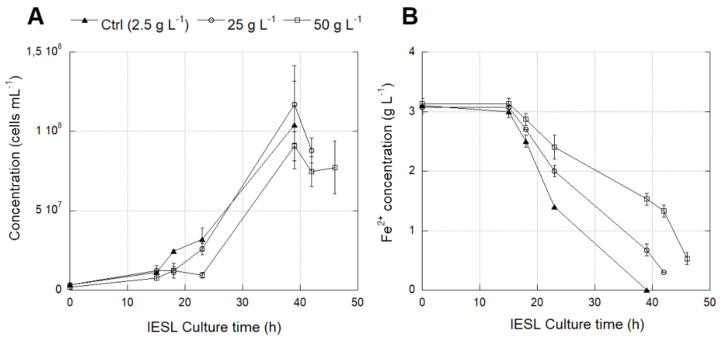
Cell growth (**A**) and Fe^2+^ consumption (**B**) of *L. ferriphilum* IESL-25 cultured under different SO_4_^2−^ concentrations (g L^−1^). Vertical bars represent the standard deviation (SD) calculated with biological replicates.

**Figure 9 genes-12-00474-f009:**
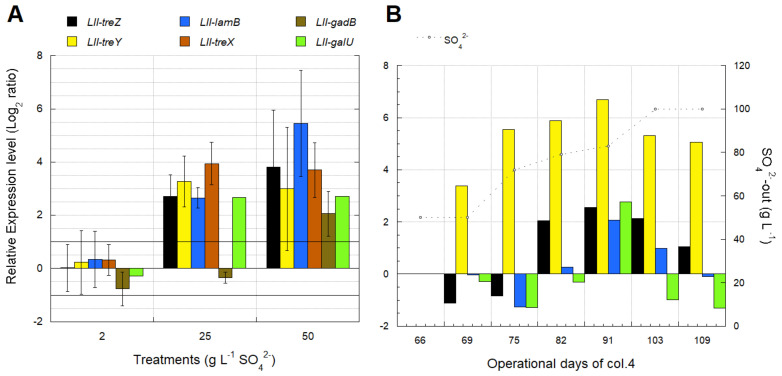
Transcriptional dynamics of trehalose synthesis genes of *L. ferriphilum* (LII) in response to (**A**) three different SO_4_^2−^ levels in pure cultures of the type strain DSM 14647, using as control one biological replicate of the cultures with 2 g L^−1^ SO_4_^2−^, and (**B**) variations in the SO_4_^2−^ concentration produced by two increments of feeding SO_4_^2−^ at operation days 68 and 103 in a bioleaching column, respectively, using as control the sample taken on operation day 66. Vertical bars represent the standard deviation (SD) calculated with biological replicates. Significant differences in relative expression were adjusted at −1 < Log_2_ ratio < 1 (horizontal black lines). In (**B**), the *X*-axis corresponds to operation days at the sampling time of col.4.

**Figure 10 genes-12-00474-f010:**
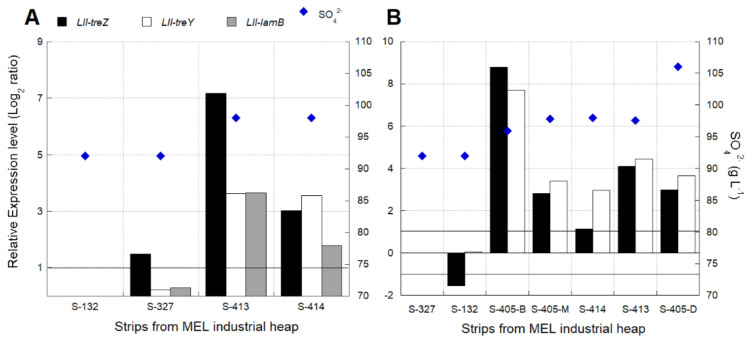
Transcriptional dynamics of *treZ*, *treY*, and *lamB* genes of *L. ferriphilum* (LII) from industrial bioleaching strips with different operational times (Table 2) and sampling (**A**) at the same date (23 August 2017) or (**B**) at different dates (11 November 2015, and 23 August 2017, Table 2). The gene *lamb* was not amplified in S-405-D (from drop), S-405-B (from the base of the second lift), and S-405-M (from the base of the fourth lift) (Table 2). Significant differences in relative expression were adjusted at −1 < Log_2_ ratio < 1 (horizontal black lines). Gene expression was calculated relative to the control samples (log 0 values) that correspond in (**A**,**B**), respectively, to S-132 and S-327.

**Figure 11 genes-12-00474-f011:**
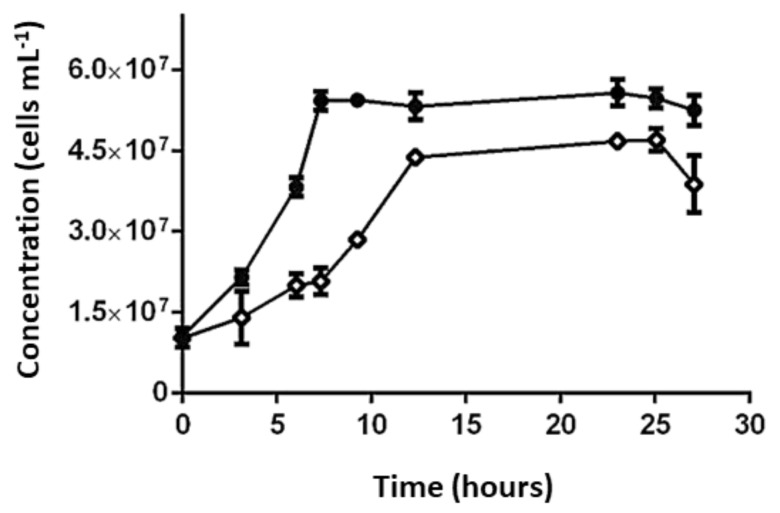
Cell growth of *Sulfobacillus* sp. CBAR13 strain growing with iron (Fe^2+^) (black circles) and tetrathionate (white circles) as the energy source. Vertical bars represent the standard deviation (SD) calculated with biological replicates.

**Figure 12 genes-12-00474-f012:**
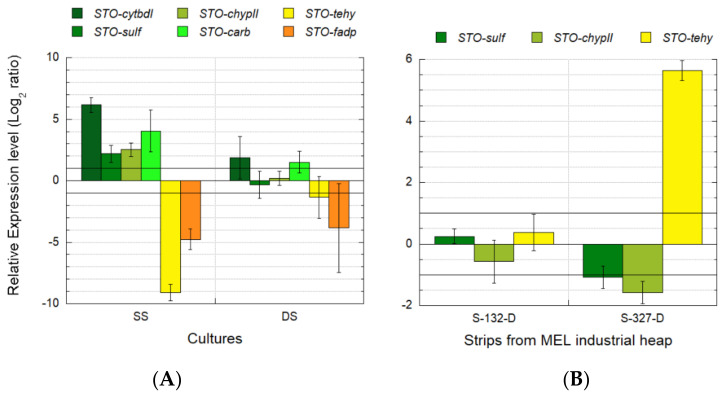
Transcriptional dynamics of *cytbdI*, *sulf*, *chypII*, *carb*, *tehy*, and *fadp* genes from *Sulfobacillus* sp. CBAR13 (STO) growing in a pure culture test with different electron donors (SS with Fe^2+^ and DS with tetrathionate) (**A**), and industrial bioleaching strips from the MEL-heap with different availabilities of Fe^2+^ (0.24 and 0.04 g L^−1^ Fe^2+^ in S-132-D and S-327-D, respectively) (**B**). Significant differences in relative expression were adjusted at −1 < Log_2_ ratio < 1 (horizontal black lines). Gene expression was calculated relative to a control sample (log 0 values) that correspond in (**A**,**B**), respectively, to one biological triplicate from DS cultures and one technical triplicate from S-132. Vertical bars in each column correspond to the standard deviation (SD) calculated with biological and technical replicates.

**Figure 13 genes-12-00474-f013:**
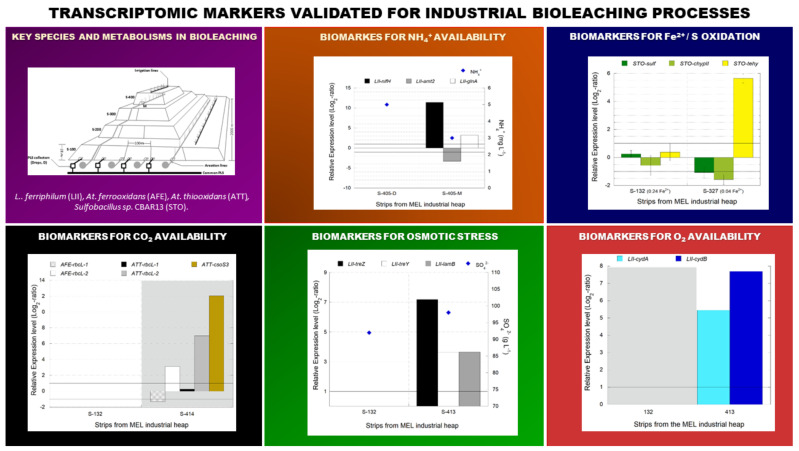
Reverse transcription quantitative PCR (RT-qPCR) array of industrially validated transcriptomic markers for monitoring copper bioleaching systems.

**Table 1 genes-12-00474-t001:** Samples collected from *L. ferriphilum* DSM 14647, *L. ferriphilum* IESL-25, *At. thiooxidans* IESL-33, and *Sulfobacillus* sp. CBAR13 cultures and from the bioleaching columns (col.1, col.2, col.3, col.4) at several operational times and specific running conditions (CO_2_, SO_4_^2−^, and NH_4_^+^ availability). (*) Air displacement with N_2_ in col.2. (**) Restoration of regular aeration in col.2. OpD, operation day; CO_2_-out, output CO_2_ concentration; SS, single-strength medium (with Fe^2+^ as an energy source), DS, double-strength medium (with K_2_S_4_O_6_ as energy source); *Sb.* sp., *Sulfobacillus* sp.

Cultures and Columns Conditions	Identification Label	SO_4_^2−^ (g L^−1^) NH_4_^+^ (mg L^−1^) Feeding	Air Feeding
*Sb.* sp. CBAR13 cultures for markers identification associated with ferrous and sulfur oxidation.	SS	-	Atmospheric
	DS	-	Atmospheric
*L. ferriphilum* DSM 14647 cultures for markers identification associated with osmotic stress (SO_4_^2−^ impurity)	LII-DSM-2	(2.5)	Atmospheric
	LII-DSM-25	(25)	Atmospheric
	LII-DSM-50	(50)	Atmospheric
*L. ferriphilum* IESL-25 cultures study of osmotic stress (SO_4_^2−^ impurity)	LII-IESL_2	(2.5)	Atmospheric
	LII-IESL_25	(25)	Atmospheric
	LII-IESL_50	(50)	Atmospheric
*At. thiooxidans* IESL-33 cultures for markers identification associated with CO_2_ availability	ATH-50	-	50 ppm CO_2_-in
	ATH-100	-	100 ppm CO_2_-in
	ATH-300	-	300 ppm CO_2_-in
	ATH-500	-	500 ppm CO_2_-in
col.1: bioleaching column #1, control	OpD60	-	200 ppm CO_2_-out
col.2: bioleaching column #2 for markers identification associated with O_2_ and CO_2_ availability	(*) OpD62	-	Air ON (500 ppm CO_2_-out)
	OpD64	-	Air OFF (2000 ppm CO_2_-out)
	(**) OpD69	-	Air OFF (1600 ppm CO_2_-out)
	OpD76	-	Air ON (200 ppm CO_2_-out)
	OpD97	-	Air OFF (200 ppm CO_2_-out)
col.3: bioleaching column #3 for markers identification associated with N_2_ and NH_4_^+^ availability (nutrients)	OpD62	(0)	Atmospheric
	OpD69	(40)	Atmospheric
	OpD76	(40)	Atmospheric
	OpD90	(40)	Atmospheric
	OpD104	(0)	Atmospheric
col.4: bioleaching column #4 for markers identification associated with osmotic stress (SO_4_^2−^ impurity)	OpD67	(50)	Atmospheric
	OpD70	(50)	Atmospheric
	OpD77	(80)	Atmospheric
	OpD84	(80)	Atmospheric
	OpD93	(80)	Atmospheric
	OpD105	(10)	Atmospheric
	OpD111	(100)	Atmospheric

**Table 2 genes-12-00474-t002:** Physicochemical characterization of industrial samples from the Minera Escondida Ltd. (MEL) bioleaching heap (strips S-405, S-410, S-132, Scheme 327. S-413, and S-414) sampled at different operation times for gene expression analysis associated with CO_2_ availability, osmotic stress, nitrogen availability, and ferrous/sulfur oxidation. Gray areas mark the operational factors and their related metabolisms studied in the respective industrial samples. Industrial strips were sampled on: ^(1)^ 24 November 2015; ^(2)^ 29 December 2016; ^(3)^ 12 January 2017; ^(4)^ 26 January 2017; ^(5)^ 23 August 2017. The marks W0, W2, and W4 in S-410 correspond to three samplings performed biweekly before (0), after 2 weeks, and after 4 weeks since the blower was turned off. D represents samples collected from the heap base, M represents samples collected at the base of the fourth lift, and B represents a sample collected at the base of the second lift of the heap. L1 and L4 represent blowers present in the heap-base or the fourth lift base, respectively. ON and OFF are blowers turned on and turned off, respectively. Day refers to the strip operation day.

Sample	Day	Heap Lift	Total Fe (g L^−1^)	Fe^2+^ (g L^−1^)	H_2_SO_4_ (g L^−1^)	pH	Eh (mV)	NH_4_^+^ (mg L^−1^)	SO_4_^2−^ (g L^−1^)	Blowers Status
^(1)^ S-405-D	284	4th	1.39	-	2.36	2.06	786	4.0	106	L1 ON
^(1)^ S-405-B	284	4th	0.84	-	1.57	2.28	798	5.0	96	L1 ON
^(1)^ S-405-M	284	4th	1.91	-	4.68	1.67	800	3.2	98	L1 ON
^(2)^ S410-W0-D	377	4th	1.46	<0.01	2.69	2.05	790	4.0	113	L1 and L4 ON
^(2)^ S410-W0-M	377	4th	1.95	0.24	3.67	1.84	676	2.2	122	L1 and L4 ON
^(3)^ S410-W2-D	391	4th	1.53	<0.01	3.01	2.05	765	1.8	104	L1 ON L4 OFF
^(3)^ S410-W2-M	391	4th	2.14	0.92	2.72	1.95	659	1.8	103	L1 ON L4 OFF
^(4)^ S410-W4-D	405	4th	1.59	<0.01	2.09	2.14	826	5.4	104	L1 ON L4 OFF
^(4)^ S410-W4-M	405	4th	2.33	0.64	5.91	1.85	666	5.8	104	L1 ON L4 OFF
^(5)^ S-132-D	63	1st	1.73	0.24	2.07	2.00	708	16	92	L1 ON
^(5)^ S-327-D	43	3th	0.83	0.04	1.29	2.16	720	10	92	L1 ON
^(5)^ S-413-D	394	4th	1.82	0.15	2.88	1.97	707	3.0	97	L1 and L4 ON
^(5)^ S-414-D	6	4th	1.65	0.15	2.38	2.08	769	2.0	98	L1 ON L4 OFF

**Table 3 genes-12-00474-t003:** Candidate genes analyzed for biomarkers identification and industrial validation. The CO_2_ fixation genes are from the Calvin–Benson–Basham cycle. HKG is housekeeping genes. SRC is sulfur-reduced compounds. *Sb.* sp. is *Sulfobacillus* sp. Genes that were used as endogenous control are indicated with (*).

Species	Gene	Gene Product	EC Number	Pathway
*A. thiooxidans*	*rbcL1*	RubisCo type I	4.1.1.39	CO_2_ fixation
*A. thiooxidans*	*rbcL2*	RubisCo type II	4.1.1.39	CO_2_ fixation
*A. thiooxidans*	*csoS2*	Carboxysome shell proteins	-	CO_2_ fixation
*A. thiooxidans*	*csoS3*	Carbonic anhydrase type ε	4.2.1.1	CO_2_ fixation
*A. thiooxidans*	*csoS4*	Carboxysome shell proteins	-	CO_2_ fixation
*A. thiooxidans*	*gyrA* *	DNA gyrase subunit A	5.99.1.3	HKG
*A. ferrooxidans*	*rbcL*-*1*	RubisCo type I	4.1.1.39	CO_2_ fixation
*A. ferrooxidans*	*rbcL*-*2*	RubisCo type II	4.1.1.39	CO_2_ fixation
*A. ferrooxidans*	*Alas* *	DNA-binding transcription factor	-	HKG
*L. ferriphilum*	*treZ*	Malto-oligosyltrehalose trehalohydrolase	3.2.1.141	Trehalose synthetase
*L. ferriphilum*	*treY*	Malto-oligosyltrehalose synthase	5.4.99.15	Trehalose synthetase
*L. ferriphilum*	*treX*	Glycogen debranching enzyme	3.2.1.68	Trehalose synthetase
*L. ferriphilum*	*lamB*	Porine maltose	-	Trehalose synthetase
*L. ferriphilum*	*galU*	Glucose-1 -phosphate-UDP-pyrophosphorylase	2.7.7.9	Trehalose synthetase
*L. ferriphilum*	*gadAB*	Glutamate descarboxilase	4.1.1.15	Trehalose synthetase
*L. ferriphilum*	*nifH*	Nitrogenase	1.18.6.1	N_2_ fixation
*L. ferriphilum*	*amt2*	Ammonium transporter	-	NH_4_^+^ transporter
*L. ferriphilum*	*glnA*	Glutamine synthetase	6.3.1.2	NH_4_^+^ asimilation
*L. ferriphilum*	*glnB*	Glutamine synthetase	6.3.1.2	NH_4_^+^ asimilation
*L. ferriphilum*	*alas* *	DNA-binding transcription factor	-	HKG
*L. ferriphilum*	16S	Ribosomal RNA small subunit	-	HKG
*Sb.* sp. CBAR13	*carb*	2Fe-2S iron-sulfur cluster binding domain-containing protein	1.2.7.4	Fe^2+^ oxidation
*Sb.* sp. CBAR13	*cytbdl*	Cytochrome bd ubiquinol oxidase subunit I	-	Fe^2+^oxidation
*Sb.* sp. CBAR13	*sulf*	Hypothetical sulfocyanin	-	Fe^2+^oxidation
*Sb.* sp. CBAR13	*chypll*	Hypothetical cbb3-type cytochrome c oxidase subunit I	-	Fe^2+^oxidation
*Sb.* sp. CBAR13	*fadp*	NAD(P)/FAD-dependent oxidoreductase	-	SRC oxidation
*Sb.* sp. CBAR13	*tehy*	Tetrathionate hydrolase	3.12.1.B1	SRC oxidation
*Sb.* sp. CBAR13	*gyrA* *	DNA gyrase subunit A	5.99.1.3	HKG

## Data Availability

Metagenome data available on request.

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
