# Peer review of "From Laboratory towards Industrial Operation: Biomarkers for Acidophilic Metabolic Activity in Bioleaching Systems"

_genes, 2021, doi:10.3390/genes12040474_

Round 1
Reviewer 1 Report
The article entitled "From laboratory towards industrial operation: biomarkers for acidophilic metabolic activity in bioleaching systems" by Dr. Marin et al., Focuses on optimization studies of bioleaching processes, mainly ROM copper bioleaching. The authors have carried out studies with the aim of improving the control of the process, the microbial community that inhabits the piles of Minera Escondida Ltda .. Several piles have been monitored and laboratory experiments have been developed for the implementation of improvement strategies as well as the determination of the cellular activity inside the heaps.
It is a very interesting work, it provides new data on the parameters that must be improved for the optimization of bioleaching processes such as CO2 concentration as well as pH measurements, among others.
However, the work presents several weak points that must be improved.
An important point to improve in general is the grammar and length of the sentences for a better understanding of the writing. Some sentences are excessively long so that they end up being ambiguous and difficult to understand. We have an example in the first paragraph of the Introduction section, lines 45-50. A sentence of 5 lines that, in fact, is a paragraph, which makes it difficult to understand.
Likewise, something that affects the entire manuscript is english grammar. I would recommend revision of English.
In particular:
Line 101: DSS (Decision-making system) Since it is a crucial point for the optimization of bioleaching processes, I recommend delving into the description of the DSS system. Describe DSS system in deep.
Lines 100-104: Very long sentence that is difficult to understand.
Materials and Methods Section, 2.1 Strains and media. The description of the media and their composition (lines 117-118, 124-128, 131-134) would be better to be removed and summarized by the corresponding reference. Or describe in supplementary material where appropriate.
Section 2.4 Candidate genes selection and primer design - lines 232-235 - Ambiguous! candidate gene markers selected for microbial diversity characterization or, for metabolic status of the leaching processes? Should be better explained. Neither the scopus nor the method is clear at all.
Line 236: Oxygen reduction, CO2 and N2 fixation: Why only those? It should be included more markers as the metanogenomic data would cover more data. Few and classical markers sellected! Science interest of the paper would increase.
Section 3.2.3 Insdustrial… .. lines 349-350 because cDNA strips did not show PCR amplification. Do you know why? Should be explained.
Section 5. Conclusions: Very brief. Conclusions section should be extended as function of the good results provided in the paper.
From line 739 – 763: Supplementary Materials, authors contributions etc… no information available. General text from the journal template.
Author Response
"Please see the attachment."

Reviewer 2 Report
This manuscript describes several qPCR primer sets that were used to detect the transcriptional activity of specific marker genes representing selected microbes and their metabolic activities in a bioleaching system. The biomarkers were tested with pure cultures, column systems in lab-scale, and samples from an industrial-scale heap-leaching system. The rationale why the selected marker genes are of interest and the approach how the primers were designed are not clear, controls for primer testing are missing, and the transferability of this specific approach to other bioleaching sites was not tested, thus the interest for a broader readership beyond this individual bioleaching site is not evident. The manuscript is lengthy and contains with 12 figures and three tables too many details but the overall objectives and main messages are hard to grasp. Sometimes the text is hard to understand due to odd wording and wrong syntax/grammar. Important information such as the accession numbers of the sequences used for primer design and the author contributions are missing, also all the statements required at the end of the manuscript (marked in yellow in the text) were not completed. Taxonomic names should be applied in the correct way, spelling out genus and species names when an organism is mentioned for the first time, and abbreviating the genus name by the initial in the following. Do not use automatic hyphenation as it leads to some errors (maybe due to wrong language settings in the Word file). References were frequently cited in the wrong style in the text, i.e. not consecutively numbered, and thus cannot be found in the reference list.
Title: Why in quotation marks?
Author affiliations: Which author has affiliation 3?
Abstract: Avoid cryptic abbreviations such as ROM, or explain them. Spell out genus names of the organisms. L 13 "copper's leading technologies extraction" - what does this mean? L 16 "maximize mine with idle capacities" - what is meant? L 22 "we developed and validated marker genes" - the genes were not developed but just applied for primer design to develop markers. L 32: How can CO2 and N2 availability be assessed by genes? Isn't it a question of CO2 or N2 fixation?
Keywords: Should be informative and not redundant to the title.
Introduction:
The background information on mining technologies is too long and too detailed, while important microbiological background information is missing. The introduction should focus on the metabolic pathways and organisms relevant for the process and introduce preceding work that was done in this field (i.e. biomarkers for mining-related bioprocesses). The objectives, research questions or hypotheses of the work should be clearly stated and the rationale for the experimental approach should be explained. Here, it is not clear if this is a method paper (to introduce a new biomarker set that can be applied to bioleaching processes in general) or a research paper applying the biomarker set to answer research questions on a specific site (here, MEL site). In the first case, the transferability of the marker set to other application sites has to be demonstrated, and specificity and coverage (false-postive and false-negative results) of the primer sets (as well as PCR conditions) need to be proven based on positive and negative controls. In the second case, the research questions on the MEL site need to be stated and set into the context of the genetic markers used.
L 63: What is meant by "copper's leading technologies"?
L 65: SX/EW ??
L 73-76: Check syntax. What requires what? Reasons for what?
L 84: What is meant by "taxonomy implementation strategy"?
L 88-89: Which functional-based strategy? Which unsolved questions?
L 91-92: Check syntax
Materials and Methods
Section 2.1: Why are CO2 concentrations given in % and ppm? This is redundant information. L 123: What is IESL strain? Each strain needs to be introduced with full taxonomic name and reference (if not received from a public strain collection such as DSMZ). Strain CBAR-13 is introduced here as Sulfobacillus sp., but later it is named S. thermosulfidooxidans. What is the source for this strain and how was it identified to the species level? Give a reference or describe the isolation source and method, how the strain was identified and what is the evidence for the species assignment. The designation should be consistent throughout the text. To make this paragraph more concise, cultivation parameters such as energy source, cultivation temperature, pH, and agitation speed could be given in Table 1. L 148: Software used for the model?
Section 2.2. L 159: "opportunities" makes no sense here. L 160: Which population analyses? L 163: neither - nor. L 171/190: Use abbreviations consistently - PLS is first explained as "Percolated Leaching Solution" and later as "Pregnant Leaching Solution".
Table 1: could be improved, the structure is not very clear and the use of () and [] for concentrations of different ions in one column is confusing.
Section 2.3. What is (40) in L 187?
Figure 1: Correct "collectors", explain PLS in the figure caption and add the correct citation of the reference Cortés et al.
Table 2: The abbreviation OD for operational day is confusing as OD usually implies optical density. Just day would be sufficient. Also FeT is confusing as it reads like a chemical formula, at least T should be subscribed.
Section 2.4. The accession numbers of the genomes used for primer design need to be given. L 253: 16S rRNA is not a gene name and thus not written in italics. If I understand the method description correctly, primers were designed based on the selected four genomes only and each of the primer sets is specific for one species. This makes the general applicability of the biomarkers questionable. It can be applied to this specific site only and for each new site the whole workflow of metagenome sequencing, genome reconstruction and species-specific primer design needs to be repeated. A more generic approach would be to download all gene sequences for a specific key enzyme from the public databases, align their amino acid sequences and derive degenerate primers targeting the conserved region. This would require extensive in silico and in vitro testing of primer specificity and coverage, but the result could be of interest for the scientific community and not only for one specific site. Were the primers tested on mock communities on DNA level before application on cDNA from real communities? Was the specificity of the primers tested by sequencing PCR products? If not, how can you ensure that non-target sequences were not amplified? Did you apply control DNA without the target sequence? L 264: What is meant by repeatability and optimal efficiencies of >100%, and what is efficiency (E) given in % in the text but having numbers of 1.8-2.3 in Table 3? What does the Pearsons coefficient refer to in this context? All these parameters and the underlying concepts in primer design need to be explained.
Tables 3/4: The genes listed in Table 3 do not explain what the target function is (which enzymes and pathways). Merge Table 3 with Table 4 or switch the order of the tables so that functions and pathways are explained first. What does * mean in Table 3? If the primers were derived from gene sequences available in databases, the accession number of these genes should be included in Table 3. If not, the sequences should be submitted.
Section 2.5. Which three genes associated whith N2 fixation? I see only nifH listed in Table 4. Genes involved in NH4 assimilation are not necessarily associated with N2 fixation. When NH4 is available as nutrient, there is no need to fix N2, as this process requires a lot of ATP to be invested by the cell.
Results:
Delete section 3.1., this is not a result.
The whole results section is tedious and has many redundant parts. The single datasets on relative gene expression levels should be more aggregated and the results clearly summarized in the text. The individual graphs could go to the supplement. Instead of grouping the results according to the single metabolic functions (CO2 fixation, N2 fixation and NH4 assimilation, osmotic stress response, iron and sulfur oxidation), I suggest to group them by the level of complexity of target samples, i.e. pure cultures, samples from column experiments, industrial heap samples. The objective stated in the introduction section needs to be addressed in the results section: The authors claimed that they developed a decision making support system for knowledge transfer and recommendations to the plant operator. However, the results presented are merely descriptive, relations between gene expression levels and process parameters appear to be erratic, and implications for process optimization or control are missing.
In the diagrams illustrating the relative expression levels (Fig. 3, 4, 5, 6, 9, 10, 11), it is not clear which gene or sample the relative expression level refers to, i.e. which is the log 0 value or the control condition - this should be always stated in the figure caption. Apart from that, interpretation of these diagrams is difficult as replicates are missing (except in Fig. 9 and 11) and thus the variance is not known. In cases where error bars are shown (Fig. 9A, Fig. 11), they were calculated from biological and technical replicates, which is not correct from the statistical perspective, and the number of replicates needs to be stated.
Fig. 5 and 9B are hard to read due to the slender columns.
Fig. 3B is wrong as the X axis implies to be a time axis but the scaling is not proportional.
L352-355: This conclusion is not justified.
L 378 and elsewhere: Why are the temporal expression profiles of the reference genes not shown (at least in the supplement)?
Fig. 12 is not mentioned in the text and its purpose is not clear. Moreover, it shows NH4 availability as target function while the text is about N2 availability and the actual target function of the nifH biomarker is N2 fixation.
Discussion:
The discussion is widely redundant to the results section and its structure is not adequate to synthesize the results and extract the key message(s).
L 549: What has an elevated CO2 concentration to do with anaerobic metabolism? Anaerobic means without oxygen, no matter what the CO2 concentration is.
L606-608 and L 617-619: Redundant information.
L 623 and 659: what is λ-aminobutyrate? Do you mean γ-aminobutyrate (GABA)?
L 727: why "meanwhile"?
Conclusions: The last sentence is not valid - I do not see how these markers were proved here to be useful.
References: correct style, ref. 28 and 33 are identical.
Supplement:
Table S1 is not relevant.
Table S2 is not readable.
Figure S2: What does "on the down" refer to? The Y axis should be designated as "Relative abundance" without % when the scale is labelled in %.
Figure S4: What is "star of irrigation"?
Author Response
"Please see the attachment."

Reviewer 3 Report
I am asking for a better elaboration of the summary. Considering such extensive work and the number of results, the summary seems to be poor, especially in terms of practical application. The work is of very good quality, the results are fully described and discussed.
Author Response
"Please see the attachment."

Reviewer 4 Report
Dear Editor and Authors,
In the manuscript presented to me for review, Marin et al have attempted to identify marker genes of common bacterial biomining species, that can serve to indicate chemical parameters that are deviating from the (for the bacteria) optimal range, and therefore may show up ways to improve the efficiency of a given biomining process. Marker genes are first discovered by metagenomics, then tested for feasibility via qPCR of pure culture reactors, column experiments, or a combination thereof, and then validated using samples collected from a operational large scale bioleaching heaps.
While I consider the study to be scientifically sound, I have major problems with the presentation of the methodology and collected data. Two features make the manuscript difficult to understand and raise many questions:
A) Figures are inconsistent in regard to colors, legends/symbols, inclusion of statistics, etc. My suggestions to fix these issues are
-Figure 1: Include more details of the heap, in particular with the sampling in mind. I.e. try to capture the reasoning of the sampling locations. This is in fact directly tied to my point B, see below.
-Figure 2: Symbols used for 500 and 300 ppm in A and B are inconsistent. Use only ONE legend for the panels, sort by concentration
Figure 3: As before, use only one legend. Why is there no data for 300 ppm in A? You are applying a discrete value (genes) to a continuous scale (days), e.g. it looks like the first black bar was measured on a different day than the grey bar. This is an issue in most of the following plots, although in some of the other plots you seem to try to mitigate by shrinking the width of the bars (e.g. Fig.5). To solve this for Figure3, I suggest to remove the line graph of carbon concentration, and instead use the carbon concentration as discrete values on the x-axis (just like in panel A of the same Figure).
Figure 4: This is again an issue that corresponds with my point B below. The sample names (S-410-M0,...) do not convey any information to the reader. I suggest using the measurement that lead to these samples being chosen as worthy for validation.
Figure 5: As above, discrete values on a continuous scale, even though it is now less obvious due the the decreased width of the bars. The decreased width makes it very hard to read. As above, I suggest using the actual measured NH4+ concentrations as values on the x-axis, or separate the figure into two panels, one with the NH4+ plot and markings when samples were taken, and the other panel with the gene expression bar plots.
Figure 6 : Redundant legends, inconsistent axis on A and B (both values and left/right), and the above mentioned issue with sample names instead of values on the x-axis (i.e. in this plot, x-axis could simply be the value marked by the NH4+ point of the same column).
Figure 7: Panel A has two controls although they consist of biological replicates. Why were both included, or why were they not summarized into one line? Also, redundant legends.
Figure 8: Redundant legends.
Figure 9: Panel A is the first gene expression bar plot that depicts standard deviations. Why were they not included in other plots? Panel B, see comments for Figure 5. Also, redundant legends.
Figure 10: See comments Figure 6.
Figure 11: See comments Figure 4.
Figure 12: This Figure is not mentioned in the text.
B) Complicated designation of the samples originating from the heap that includes "drops", "bases", "lifts", "levels", abbreviated and indexed with numbers, etc. These sample designations do not actually convey any information to the reader about how and why they are utilized as a sample in which context. For example, I can not retrace/comprehend why in Figure 4 Samples S-410-M0, S-410-D2, S-410-D4, and S-410-M2 are compared as I have no understanding of these samples. Are they a time series, are they different locations of a heap that are otherwise entirely unrelated? This issue is not mitigated by inclusion of Table 2, as it is not feasible to review every sample code there, and Table 2 is crowded and lacks information about what the sample is actually supposed to represent. I suggest to either rename all samples in a way that allows the reader to gain some information about the rationale (or function) of why this sample was used (not only the location), or to first and foremost refer to the sample by the important measurement instead of the name (e.g. the CO2 concentration when talking about samples used to compare gene expression of CO2 related genes).
Beside these two major issues, I include my recommendations about several smaller ones:
line 11: The abbreviation "ROM" has not been defined yet.
line 34: "MEL" has not been defined yet.
line 45-50: This sentence is not easily comprehensible, I suggest dividing it.
line 52-53: "because of its kinetics and recovery", please clarify.
line 65: "SX/EW" has not been defined yet. First main text occurrence of "ROM"
line 85: Which approaches are meant? If the ones in the previous sentence, just use "these approaches" instead of "approaches mentioned above".
line 87-90: Please clarify.
line 93: Seems something is missing
Methods:
-please include how CO2 was supplied (bubbling? carbonate?)
-why were two Leptospirillum ferriphilum strains chosen?
-line 125 and 127: MgSO4x7H2O is specified twice
-line 130: Here, Sulfobacillus is referred to as "Sulfobacillus sp. strain CBAR-13", in the rest of the manuscript as "Sulfobacillus thermosulfidooxidans CBAR-13"
-was Sulfobacillus grown without addition of organic carbon?
line 139: What about Sulfobacillus' cell growth?
line 139-140: Please provide description of the procedure or a reference.
line 148: Please provide a reference for the model.
line 151: Please describe the open circuit system
line 152: What is the chemical composition of the raffinate and what is the flow rate?
line 171 and 190: redundant definitions of PLS
Table 1: Please replace "Tests" with parameters such as "Organisms", "Target parameter", or similar.
line 236: Why was At. ferrooxidans not included in the study?
line 245: Why is Lycantay of relevance?
line 257-263: Please explain why this was undertaken
Table 3: I suggest this to be moved to the Supplementary information.
line 310: Please add "as assessed by cell counts" after "growth rates of At. thiooxidans IESL-33"
line 324-331: Why is there no data on 300ppm CO2?
line 347: It is my understanding that you attempted to validate (for industrial large scale use) the marker genes predicted by pure culture and column experiments. E.g. in section 3.2.3, you attempt to validate the use of CO2 fixation genes to reveal CO2 depletion. You amplify carbon fixation genes obtained from the heap, find a higher than normal occurrence of these genes in some samples, and conclude that there is a low availability of CO2 (line 357) in the area where those samples were taken. However, you do not appear to present a physical measurement of the CO2 concentration at this location of the heap, which in my opinion is the actual validation necessary to judge if the method works.
line 393-394: See issue B, what was the rational that allowed these "strips" to be used?
line 399: "Decrease UP to...." Something is not right here
line 410: Please explain why two strains are analyzed for L. ferriphilum
Author Response
"Please see the attachment."

Round 2
Reviewer 2 Report
Although the manuscript has been improved compared with the first version, there are still major issues with regard to the method description and data availability. Some of my previous comments were not sufficiently addressed, in particular the description of the primer design. It is still not clear which genes exactly were used as just the accession numbers of whole genome assemblies were given, for the reconstructed genomes from MEL no accession at all. The data availability statement "Metagenome data available on request" is not acceptable. If the authors do not want to publish the whole genomes, they should at least submit the annotated fasta files comprising the genomic regions with the coding sequences that were used for primer design. Apart from that, it is hard to see how the reviewers' comments were addressed since the authors did not provide line numbers of the changes they made to the text, and the line numbers of the revised version do not correspond with those of the original manuscript. The manuscript is still lengthy and with 13 figures shows too many details.
Reviewer 4 Report
Dear Authors,
thank you for addressing my previous comments. I acknowledge that a large scale biomining heap is a enormously complicated system that is challenging to describe. That is the exact reason that the UTMOST care should be given to explaining to the unfamiliar reader the rationale and intend of sampling design. In this regard, the presented manuscript could still be improved.
I also retain that, while on first sight somewhat intuitive, mixing discrete and continuous scales as in Figures 3, 5, etc is simply factually wrong. I acknowledge, however, that (regrettably) this is often seen in Manuscripts these days, and will not hold the authors to higher standards than the rest of the scientific community is willing to enforce. Nonetheless, I encourage the authors to avoid this in the future. Alternative figure designs are easy to conceive and often just as simple to comprehend.
Nevertheless, the manuscript has been improved considerably since its first submission.